# Network-Compatible Unconditionally Secured Classical Key Distribution via Quantum Superposition-Induced Deterministic Randomness

**Byoung S. Ham** 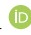

School of Electrical Engineering and Computer Science, Gwangju Institute of Science and Technology, Gwangju 61005, Korea; bham@gist.ac.kr; Tel.: +82-62-715-3502

**Abstract:** Based on the addressability of quantum superposition and its unitary transformation, a network-compatible, unconditionally secured key distribution protocol is presented for arbitrary networking in a classical regime with potential applications of one-time-pad cryptography. The network capability is due to the addressable unitary transformation between arbitrary point-to-point connections in a network through commonly shared double transmission channels. The unconditional security is due to address-sensitive eavesdropping randomness via network authentication. The proposed protocol may offer a solid platform of unconditionally secured classical cryptography for mass-data communications in a conventional network, which would be otherwise impossible.

**Keywords:** cryptography; unconditional; quantum superposition; optical network compatible





## 1. Introduction

Due to the exponential growth of information traffic in fiber-optic communications backbone networks over the last thirty years, the information traffic rate has tripled every two years and is expected to reach its theoretical upper bound of 100 Tbps within a decade [1]. The more information traffic increases, the more data security should be emphasized. Current information security relies on computational complexity [2] and is thus vulnerable to both classical [3] and quantum attacks [4–6]. In classical cryptography such as public key cryptography [7], the key length has gradually increased over decades to protect data from potential eavesdropping, mostly relying on computing power [8]. As a result, secured data transmission in a classical (unsecured) regime becomes inefficient as the key length increases due to the tradeoff between security and the key generation rate [8]. Especially for big data-based artificial intelligence applications such as unmanned vehicles and Internet of things applications such as drones, data security must be carried out in an efficient way [9]. Thus, fundamental innovation in cryptography is required to overcome vulnerabilities in both classical attacks relying on algorithms or computing powers [3] and quantum attacks relying on quantum parallelism of superposition [4].

On the contrary, quantum cryptography [10] has been intensively studied for unconditionally secured quantum key distribution (QKD) over a quantum channel ever since the first QKD protocol of BB84 [11]. Due to imperfect single-photon detectors and quantum channel losses resulting in quantum loopholes however, QKD is also vulnerable to quantum attacks from a practical point of view [12]. The detection loopholes affect all QKD protocols, including decoy states [13] for single photons and Bell states [14] for entangled photon pairs. For transmission distance, QKD is strongly limited by the no-cloning theorem prohibiting duplication or amplification [15], unless quantum repeaters are implemented [16]. Moreover, there are no commercially available deterministic single-photon or entangled-photon pair generators yet, resulting in an extremely low QKD rate [10]. Besides, the key must be used only once to keep the unconditional security guaranteed by quantum mechanics [17]. Quantum networking among many parties is much harder to realize due to the limitations

of multipartite entangled photon-pair generation [18]. Based on these practical issues, quantum cryptography seems to have a long way to go for commercial network applications such as e-commerce, including online banking and IoT via both wired and wireless communications [19], even though some point-to-point QKD protocols have already been launched for a testbed [20,21]. Further, QKD is incompatible with conventional information infrastructures in the classical domain such as wired and wireless networks, and thus severely limits its applications in mass data communications such as artificial intelligence based on big data [22].

To overcome the limitations of classical and quantum cryptographies, an entirely different method of unconditionally secured classical key distribution (USCKD) has been proposed for both wired [23] and wireless [24] transmissions using a pair of transmission channels forming a Mach-Zehnder interferometer (MZI) via quantum superposition between the MZI channels and its unitary transformation, resulting in deterministic randomness.This deterministic randomness represents no eavesdropping due to measurement indistinguishability caused by quantum superposition in the MZI channels, as well as the deterministic key distribution between two remote parties via unitary transformation. As demonstrated, the key generation determinacy in USCKD [23] is well understood in the coherence optics of MZI in terms of the directional determinacy [25]. The basis of eavesdropping randomness in USCKD has also been understood as measurement indistinguishability caused by channel superposition, as in Young's double-slit experiments [26]. Here, a network-compatible USCKD (NC-USCKD) protocol is presented, analyzed, and discussed for arbitrary networking in the classical domain, where a commonly shared pair of transmission lines of MZI plays a key role in both the physics and infrastructure. The classical channel represents a lossy and unsecured transmission line, resulting in open access by anyone. In the proposed NC-USCKD scheme, unconditional security is achieved coherently via addressable quantum superposition between two arbitrary parties in a network through the shared MZI channels. For the robustness of the MZI system, real-time phase stabilization has already been experimentally demonstrated for a few km ranges in both wired [27] and wireless schemes [28].

For the network addressability of the present NC-USCKD, addressable quantum superposition between arbitrary two-remote parties is presented as a building block of unconditionally secured classical networking. Compared with the original point-to-point transmission scheme of USCKD [23], the addressability in the present NC-USCKD is due to the linear expansion of orthogonal bases through the shared MZI channels for N-to-N networking. For the unconditional security of NC-USCKD in a classical network, we also present an authentication protocol via network initialization between any arbitrary parties. The practical advantages of NC-USCKD include high-speed key distribution, addressable networking, and compatibility with conventional optical systems relying on the wave nature of coherence optics. Owing to the coherence optics of MZI [25,26], NC-USCKD is naturally compatible with classical systems such as optical switches, optical routers and even optical amplifiers. The phase locking in an optical amplifier such as an erbium-doped fiber amplifier is technically assured due to its coherence optics for regeneration in the fiber-optic communications networks [29]. The classical compatibility offers a great benefit to the current bottlenecked big-data applications based on CMOS technologies and can lead to a breakthrough in present mass data communications networks.

## 2. Materials and Methods

Numerical calculations in the results are conducted by homemade program using MATLAB, where the equations are driven in the main text analytically.

## 3. Results

Figure 1 shows a schematic of the proposed NC-USCKD based on a shared pair of round-trip MZI transmission channels in an N-party composed network, where addressable remote parties are called Alice and Bob. Here, the round trip configuration of MZI is the

same as symmetrically coupled double MZIs, where Bob (Alice) controls the first (second) MZI. For the N-party networks, the number of arbitrary pairs for networking is $\frac{N(N-1)}{2}$, which is a quadratic expansion. This quadratic scalability in networking may be solved via multi-party superposition, which is beyond the present scope (discussed elsewhere). Each party has dual phase shifters to encode/encode one's phase bases represented by, for example, $\varphi_1$ and $\varphi_2$ for the phase shifters $\Phi_1$ and $\Phi_2$ at Bob's side or $\psi_1$ and $\psi_2$ for $\Psi_1$ and $\Psi_2$ at Alice's side, respectively. The MZI scheme in Figure 1 has nothing to do with the phase encoded BB84 protocol [30], where USCKD uses a pair of transmission channels for deterministic randomness via quantum superposition and its unitary transformation [23]. For NC-USCKD, the phase controllers $\Phi_2$ and $\Psi_2$ are added to the original scheme of USCKD for the purpose of addressable networking, where the original phase controllers ($\Phi_1$ and $\Psi_1$) are used for the unconditional security via deterministic randomness in the doubly coupled MZIs. In USCC without $\Phi_2$ and $\Psi_2$ [23], the round-trip MZI results in the deterministic randomness if $\varphi_1 = \psi_1$ is satisfied, where $\varphi_1$ and $\psi_1$ have the same set of orthogonal phase bases: $\varphi_1, \psi_1 \in \{0, \pi\}$. The opposite case of $\varphi_1 \neq \psi_1$ also works for the key distribution if bit-by-bit network initialization is performed [23]. Here, we briefly seek an N-party addressable condition in the NC-USCKD scheme of Figure 1: The phase basis '0' ('$\pi$') represents the key '0' ('1').

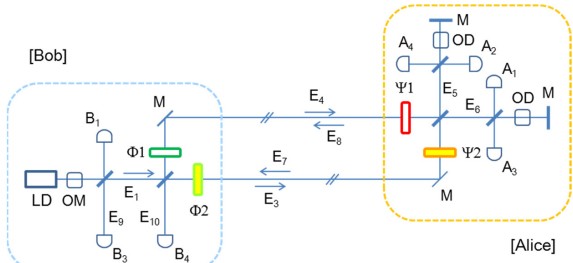

**Figure 1.** A schematic of NC-USCC. LD, Laser diode; OM, Optical modulator; $\Phi1$, $\Phi2$, $\Psi1$, $\Psi2$, Phase shifter; $A_1 \sim A_4$, $B_1, B_3, B_4$, Photodetector, OD, Optical delay, M, Mirror, $E_i$, Light i. The distance between Bob and Alice depends on the MZI stability which can be in the order of 10 km. For the whole network configuration, refer to Section C of the Supplementary Information.

The matrix representation, [BH], for the dual phase-controlled round-trip MZI in Figure 1 is as follows (see Section A of the Supplementary Information):

$$[\text{BH}] = -\frac{1}{2} \begin{bmatrix} \left\{ e^{i(\psi_2+\varphi_1)} + e^{i(\psi_1+\varphi_2)} \right\} & -i\left\{ e^{i(\psi_2+\varphi_1)} - e^{i(\psi_1+\varphi_2)} \right\} \\ i\left\{ e^{i(\psi_2+\varphi_1)} - e^{i(\psi_1+\varphi_2)} \right\} & \left\{ e^{i(\psi_2+\varphi_1)} + e^{i(\psi_1+\varphi_2)} \right\} \end{bmatrix}. \tag{1}$$

According to the unitary transformation in a round-trip MZI configuration of Figure 1, the returned light ($E_9$ and $E_{10}$) at Bob's side must satisfy the identity or inversion relation if no network error occurs: $\begin{bmatrix} E_9 \\ E_{10} \end{bmatrix} = [BH] \begin{bmatrix} E_1 \\ 0 \end{bmatrix}$. Here, the added phases ($\varphi_2, \psi_2$) are the assigned address parameters for their sites. As discussed already in USCKD [23], unconditional security is performed with the phase bases of $\varphi_1$ and $\psi_2$. For a fixed address set ($\varphi_2, \psi_2$), Bob randomly prepares a key with his phase basis $\varphi_1$, and sends it to Alice: this is the key preparation stage. Relative to Bob's prepared lights ($E_3, E_4$), Alice's phase ($\psi_1, \psi_2$) is transparent. Likewise, Bob's phase ($\varphi_1, \varphi_2$) is also transparent to the returned light ($E_7, E_8$). Alice measures her visibility $V_A$ to copy Bob's choice of $\varphi_1$ (see Table 1 in [23]). Then, Alice randomly chooses her phase basis for $\psi_1$ to shuffle Bob's phase choice and sends it back to Bob: this is the key selection stage. If the returned light $E_9$ ($E_{10}$) hits the detector $B_3$ ($B_4$), the identity (inversion) relation is satisfied for the unitary transformation of the MZI matrix in Figure 1 (Section A of the Supplementary Information). If Alice chooses the same (opposite) basis as Bob, this results in the identity (inversion) relation. Unlike

QKD, the key distribution of USCKD is fully deterministic without the need for sifting due to the MZI directionality, where sifting is used to induce eavesdropping randomness and the unconditional security is provided by the no-cloning theorem of quantum mechanics in QKD [15]. Thus, the random phase shuffling by Alice corresponds to sifting of the QKD for eavesdropping randomness. Here in NC-USCKD, eavesdropping randomness is achieved by network initialization (see Table 1). Depending on the key distribution strategy, the inversion case (Section B of the Supplementary Information) can also be included (see Table 2) for the key distribution. From Equation (1), the following phase relationship between Alice and Bob is obtained for identity and inversion relations, respectively:

$$\psi_1 + \varphi_2 = \psi_2 + \varphi_1, \tag{2}$$

$$(\psi_2 + \varphi_1) = (\psi_1 + \varphi_2) \pm \pi, \tag{3}$$

with a deterministic key distribution according to the MZI physics of transmission directionality, the control phase bases $(\varphi_1, \psi_1)$ in Equations (2) and (3) must be shifted by the address phase $(\varphi_2, \psi_2)$. For example, the modified phase basis $\varphi_1$ in Figure 1 is $\varphi_1 = \varphi'_1 + \psi_2$, where $\varphi'_1$ is the original binary basis $(0, \pi)$; similarly for $\psi_1$: $\psi_1 = \psi'_1 + \varphi_2$, where $\psi'_1$ is also the original binary phase basis (discussed in Figure 2). Due to the phase matching condition of $\psi_1 = \varphi_1$ $(\psi_1 = \varphi_1)$ in USCKD for the identity relation, Equation (2) results in $\psi_2 = \varphi_2$ $(\psi_2 = \varphi_2 \pm \pi)$. In a similar analogy for the inversion case of $\psi_1 \neq \varphi_1$ $(\psi_1 = \varphi_1 \pm \pi)$, the modified phase basis becomes $\psi_2 \neq \varphi_2$ $(\psi_2 = \varphi_2 \pm \pi)$ (Section B of the Supplementary Information). Owing to the network addressability with $\psi_2 = \varphi_2$ or $\psi_2 = \varphi_2 \pm \pi$ for the identity or inversion case, NC-USCKD works for any arbitrary phase address. Thus, Figure 1 functions as a basic building block of network compatible USCKD in the classical domain.

**Table 1.** Network initialization for Table 2.

| Party | Order (N) Sequence | | 1 | 2 | 3 | 4 | 5 | 6 | 7 | 8 | 9 | 10 |
|-------|------|------|-----|-----|-----|-----|-----|-----|-----|-----|-----|-----|
| Alice | 2 | $V_A$ | 1 | −1 | −1 | 1 | −1 | 1 | 1 | 1 | −1 | 1 |
| | | $\psi$ | $\delta$ | $\delta$ | $\delta + \pi$ | $\delta$ | $\delta + \pi$ | $\delta + \pi$ | $\delta$ | $\delta + \pi$ | $\delta$ | $\delta + \pi$ |
| | 4 | Correctness | X | O | X | X | X | O | X | O | O | O |
| Bob | 1 | $\varphi$ | 0 | $\pi$ | $\pi$ | 0 | $\pi$ | 0 | 0 | 0 | $\pi$ | 0 |
| | 3 | $V_B$ | +1 | −1 | +1 | +1 | +1 | −1 | +1 | −1 | −1 | −1 |

$V_A = V_{5,6}$; $V_B = V_{9,10}$. Table 1 is for $\pi$—added $\delta$. For non-$\pi$—added $\delta$, see [23]. The order number is to show random cases. "O" ("X") represents a correct (wrong) one.

In more detail, the control phase $\varphi_1$ depends on $\psi_2(= \varphi_2)$ for arbitrary networking with a particular address $\delta$ at $\Phi_2$, where $\varphi_1 = \varphi'_1 + \psi_2(\delta)$. Obviously, the $\varphi_1$ value varies based on the assigned address with $\delta$ at $\Phi_2$. As a result, the corresponding phase $\psi_1$ at Alice's side also becomes shifted by $\delta$, satisfying Equation (2), resulting in $\varphi_1 = \psi_1$ for the identity relation; otherwise, $\varphi_1 = \psi_1 \pm \pi$ for the inversion relation. Here, the address phase $\delta$ at $\Phi_2$ plays a key role in addressable networking in NC-USCKD, where $\delta$ can be considered as a continuous phase variable (CPV). This is the generalization of USCKD for networking without changing the original physics of USCKD. Keeping this in mind, we investigate the CPV property in NC-USCKD for the network addressability.

**Table 2.** A key distribution procedure for NC-USCKD in Figure 1. The phase $\varphi_1$ is denoted without addition of $\varphi_2$ for simplicity. So does $\psi_1$. The red indicates a network error. Each 'order' needs the network initialization in Table 1, otherwise sifting for the identity relation is needed.

| Party | Order / Sequence | | 1 | 2 | 3 | 4 | 5 | 6 | 7 | 8 | 9 | 10 | set |
|---|---|---|---|---|---|---|---|---|---|---|---|---|---|
| Bob | 1 | $\varphi_1$ | 0 | 0 | $\pi$ | 0 | $\pi$ | $\pi$ | 0 | $\pi$ | 0 | $\pi$ | |
| | 2 | Prepared key: $x(\varphi_1)$ | 0 | 0 | 1 | 0 | 1 | 1 | 0 | 1 | 0 | 1 | $\{x\}$ |
| | 8 | $V_B$ | 1 | −1 | 0.9 | 1 | −1 | −1 | −1 | 1 | 1 | 1 | |
| | 9 | Raw key | **0** | **1** | **X** | **0** | **1** | **1** | **1** | **0** | **0** | **0** | $\{m_B\}$ |
| | 10 | Final key | **0** | **1** | **X** | **0** | **1** | **X** | **1** | **0** | **0** | **0** | $\{m\}$ |
| Alice | 3 | $V_A$ | 1 | 1 | −1 | 1 | −1 | −0.8 | 1 | −1 | 1 | −1 | |
| | 4 | Copy x: y | 0 | 0 | 1 | 0 | 1 | −0.8 | 0 | 1 | 0 | 1 | $\{y\}$ |
| | 5 | $\psi_1$ | $\pi$ | 0 | 0 | $\pi$ | $\pi$ | $\pi$ | 0 | 0 | $\pi$ | 0 | |
| | 6 | $z(\psi_1)$ | 1 | 0 | 0 | 1 | 1 | 1 | 0 | 0 | 1 | 0 | $\{z\}$ |
| | 7 | Raw key | **0** | **1** | **0** | **0** | **1** | **X** | **1** | **0** | **0** | **0** | $\{m_A\}$ |
| | 10 | Final key | **0** | **1** | **X** | **0** | **1** | **X** | **1** | **0** | **0** | **0** | $\{m\}$ |

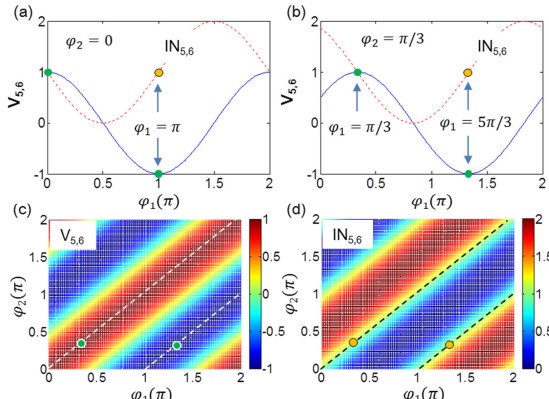

**Figure 2.** Numerical calculations for the transmission directionality in MZI. Visibility $V_{5,6}$ (solid) and Interference $IN_{5,6}$ (dotted) for (**a**) $\varphi_2 = 0$ and (**b**) $\varphi_2 = \pi/3$. (**c**) $V_{5,6}$ and (**d**) $IN_{5,6}$. $V_{i,j} = \left(\frac{I_j - I_i}{I_i + I_j}\right)$, where $I_i$ is intensity of $E_i$. $IN_{5,6} = (E_5 + E_6)(E_5 + E_6)^*$.

Figure 1 shows a paired party assigned to the address set $(\varphi_2, \psi_2)$ through a shared pair of transmission channels of MZI in the N-party network (Section C of the Supplementary Information). The coherent (bright) input light pulse $E_1$ in Figure 1 is launched from a coherent laser (LD) through an optical modulator (OM) by Bob. A random phase basis $\varphi_1 \in \{0, \pi\}$ controlled by the phase shifter $\Phi_1$ is added to the split light $E_4$. The other split light $E_3$ is encoded by the address phase shifter $\Phi_2$ with a phase variable $\varphi_2$, where $0 \leq \varphi_2 \leq \pi$. As explained above, only the $\varphi_2$—corresponding receiver (Alice) with the $\psi_2$ address satisfies Equations (2) and (3) for deterministic randomness of USCKD through the commonly shared pair of MZI transmission channels. Here, the MZI determinacy represents the phase-dependent transmission directionality: If $\varphi_1 = 0$ ($\varphi_1 = \pi$) assuming no network errors, detector $A_1$ ($A_2$) always clicks with $E_6$ ($E_5$) for $\varphi_2 = \psi_2 = 0$. The $\psi_1$—controlled returned light $E_8$ along with $E_7$ by Alice is also governed by the same MZI transmission directionality, resulting in the identity or inversion relation (discussed in Figures 2 and 3). For the return lights of $E_7$ and $E_8$, both phases $\varphi_1$ and $\varphi_2$ are invisible as mentioned above. Likewise, $\psi_1$ and $\psi_2$ are invisible to $E_3$ and $E_4$, respectively.

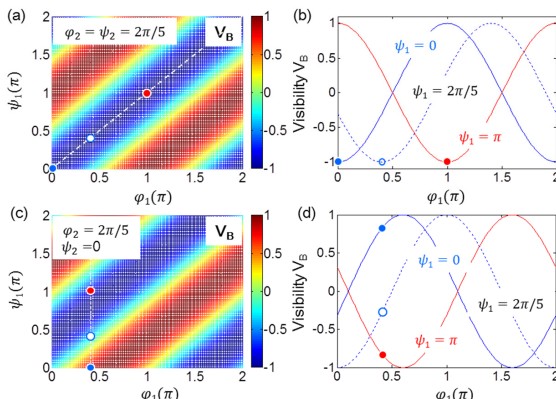

**Figure 3.** Numerical calculations of visibility $V_B$ for OKD. Visibility $V_B$ (**a,b**) for $\psi_2 = \varphi_2 = 2\pi/5$, and (**c,d**) for $\psi_2 = 0$ and $\varphi_2 = 2\pi/5$. Calculations are based on Equation (1). $V_B = \left( \frac{I_{10} - I_9}{I_{10} + I_9} \right)$: $I_i$ is the intensity of $E_i$.

Figure 2 shows numerical calculations of the MZI determinacy for the output lights $E_5$ and $E_6$ on Alice's side as well as the measurement randomness ($IN_{5,6}$) in the shared pair of transmission channels. The related matrix representation $[MZ]_{\varphi_1,\varphi_2}$ of the directionality for $E_5$ and $E_6$ at the MZI interferometer is as follows:

$$[MZ]_{\varphi_1,\varphi_2} = \frac{1}{2} \begin{bmatrix} e^{i\varphi_2} - e^{i\varphi_1} & i\left(e^{i\varphi_2} + e^{i\varphi_1}\right) \\ i\left(e^{i\varphi_2} + e^{i\varphi_1}\right) & -\left(e^{i\varphi_2} - e^{i\varphi_1}\right) \end{bmatrix}, \tag{4}$$

where $\begin{bmatrix} E_5 \\ E_6 \end{bmatrix} = [MZ]_{\varphi_1,\varphi_2} \begin{bmatrix} E_1 \\ 0 \end{bmatrix}$. The added phase $\varphi_2(\delta)$ causes a $\delta$—phase shift in $E_3$ in the lower transmission line. To compensate the phase shift, $\varphi_1$ must be adjusted accordingly for $E_4$ in the upper transmission line. Thus, the modified phase at $\Phi_1$ must be $\varphi_1 = \varphi_1' + \delta$, where $\psi_2 = \delta$, $0 \leq \delta \leq \pi$, and $\varphi_1'$ is the binary phase basis of $\{0, \pi\}$. With this modified phase, Equation (4) can be easily proved for the MZI determinacy (directionality) with an arbitrary value of $\delta$ for $\varphi_2$.

For the numerical demonstrations of the $\varphi_2$—dependent MZI determinacy mentioned above, two basis values of $\varphi_1' \in \{0, \pi\}$ are used to test both the visibility $V_{5,6}$ and the interference $IN_{5,6}$. Here, the interference $IN_{5,6}$ should be the same as $IN_{3,4}$ if Eve has the same measurement tool as Alice's. However, Eve's measurement with the same interference tool results in either an in-phase or out-of-phase scenario with the same probability due to the measurement indistinguishability caused by the MZI path superposition. Figure 2a is the reference for $\varphi_2 = 0$, while Figure 2b is for any arbitrary value of $\varphi_2 = \pi/3$. Figure 2a shows a typical fringe pattern of visibility $V_{5,6}$, where the maximum occurs at the phase bases, $\varphi_1' = \varphi_1 \in \{0, \pi\}$ (see the green dots in the solid curve). On the contrary, the interference $IN_{5,6}$ results in the same value for both bases, resulting in measurement indistinguishability (see the green and orange dots in the dotted curve). As discussed in [21], $IN_{5,6}$ should be the same as $IN_{3,4}$, showing the physical origin of the measurement immunity in the MZI path corresponding to the no-cloning theorem in QKD. The phase shift of $\varphi_1$ by the address value of $\varphi_2$ is numerically demonstrated in Figure 2b for $\varphi_1 = \varphi_1' + \varphi_2\left(\frac{\pi}{3}\right)$. For the maximum visibility $V_{5,6} = \pm 1$, the phase shift condition is also satisfied. This linear phase shift relation in $\varphi_1$ with $\varphi_2$ reveals the infinite number of phase variables in $\varphi_2$, resulting in the CPV characteristics of the present protocol as shown in Figure 2c. In other words, the address phase $\varphi_2$ is used for networking to the corresponding $\psi_2$ at Alice's side. The corresponding interference $IN_{5,6}$ always has the same value if $\varphi_1 = \varphi_1' + \varphi_2$ is satisfied, as shown in Figure 2d. Thus, Figure 2 demonstrates the $\varphi_2$—dependent MZI directionality in the NA-USCKD scheme of Figure 1 as well as the indistinguishability in eavesdropping (discussed later). The resulting addressable condition on Alice's side is $\psi_1 = \psi_1' + \psi_2$.

Because the relation $\varphi_1 = \psi_1$ must be satisfied for the one-way deterministic key transmission in Figure 1, $\psi_2$ on Alice's side must be equal to $\varphi_2$ according to Equation (2). Figure 3 shows the numerical calculations for the present NA-USCKD with addressable CPV of $\varphi_2$ and $\psi_2$. To satisfy the identity matrix at Bob's side for the returned light, the visibility of $V_B = -1$ for both bases $(\varphi_1 = \psi_1 = \{\varphi_2, \pi + \varphi_2\})$ is numerically shown in Figure 3a for the right condition of $\varphi_2 = \psi_2\left(\frac{2\pi}{5}\right)$: $V_B = V_{9,10}$. However, for the wrong condition of $\varphi_2 \neq \psi_2\left(\frac{2\pi}{5}\right)$, the maximum visibility of $V_B$ fails. Thus, Equations (2) and (3) are proved, where the modified phase basis of $\varphi_1$ becomes continuous because $\varphi_2(= \psi_2)$ is continuous: $0 \leq \varphi_2 \leq \pi$. In practice however, the possible number of CPV is of course determined by the detector's sensitivity and MZI phase stability.

Figure 3a,b represents for the $\psi_1$—independent identity relation $\left(\psi_1 = 0; \frac{2\pi}{5}; \pi\right)$ in the round-trip MZI scheme of Figure 1. For the address matching condition $(\varphi_2 = \psi_2)$ as shown with the dashed curve in Figure 3a, all $\psi_1$ values satisfy the correct $V_B$ if $\psi_1 = \varphi_1$. The visibility $V_A$ ($=V_{5,6}$) is broken if $\varphi_1 \neq \varphi_1' + \varphi_2$ (see Figure 2b). Thus, only the dotted curve with $\psi_1 = \frac{2\pi}{5}(=\varphi_2)$ in Figure 3b satisfies directionality condition in both sides with $V_{5,6} = -1$ and $V_B = -1$ (see the open circle). This is because $\varphi_1$ must be shifted by the $\varphi_2$ value, and the shifted $\varphi_1$ affects $\psi_1$ to keep $V_{5,6} = \pm 1$.

For the key distribution process in Figures 2 and 3, how does Alice know the correct $\psi_1$? In other words, how does Bob send his prepared key to Alice without revealing it to Eve? The answer to this question is given by authentication. If $\varphi_2 \neq \psi_2$ for a wrong choice, the identity relation $(V_B = -1)$ must fail as shown in Figure 3c,d (see the open circles). For the correct choice $(\varphi_2 = \psi_2)$, both Bob and Alice automatically have $\varphi_2$—phase shifted $\varphi_1$ and $\psi_1$, respectively. Thus, their visibility measurements must fulfill the identity (or inversion) relation. If there is any mismatch in the address $(\varphi_2 \neq \psi_2)$, the return light cannot satisfy the identity (or inversion) relation as shown in Figure 3d (see the open circle): $V_B \neq -1$. Here, $V_B \neq -1$ means that detector $B_4$ is also clicked on for $E_4$, indicating an error. Like USCKD [23], this property of NA-USCKD is also deterministic in the key distribution with random eavesdropping owing to the MZI physics. Details of authentication are discussed in the section on network initialization.

Figure 4 shows numerical calculations for the MZI channel measurements in Figure 1 for the demonstration of unconditional security in NC-USCKD. The matrix representation $[MZ]_{\psi,\varphi}$ is for both $E_7$ and $E_8$ in the MZI paths of Figure 1:

$$[MZ]_{\psi,\varphi} = \frac{1}{\sqrt{2}} \begin{bmatrix} -e^{i(\psi_2+\varphi_1)} & ie^{i(\psi_2+\varphi_1)} \\ ie^{i(\psi_1+\varphi_2)} & -e^{i(\psi_1+\varphi_2)} \end{bmatrix}, \tag{5}$$

where $\begin{bmatrix} E_7 \\ E_8 \end{bmatrix} = [MZ]_{\psi,\varphi} \begin{bmatrix} E_1 \\ 0 \end{bmatrix}$ is satisfied (see Section D of the Supplementary Information). Figure 4 shows both the interference $IN_{7,8}$ and visibility $V_{7,8}$ in the shared MZI channels for a smart eavesdropper. Although the channel intrusion by Eve without altering the output fringe is theoretically and technically possible with the same measurement tool, Eve's chance to decode is just 50% on average because there is no way to keep the same phase difference as Bob or Alice. In other words, the same fringe pattern (visibility) can be achieved by Eve, but the absolute phase information of the light carrier is impossible due to the superposition between the two paths. Thus, Eve's eavesdropping chance with fringe coincidence is random, resulting in unconditional security. Moreover, a random phase-basis selection technique is added to prevent classical attacks such as memory-based attacks [23]. According to Equation (2), Alice's phase adjustment on $\psi_1$ with $\psi_2$ is automatic as discussed in Figure 3. Figure 4a,b is for the address matching $(\varphi_2 = \psi_2)$ between Alice and Bob, while Figure 4c,d is for mismatching $(\varphi_2 \neq \psi_2)$. Regardless of knowing or unknowing the address set $(\varphi_2, \psi_2)$, Eve's channel attack must fail due to the MZI physics as well as the channel independence of coherence optics, as shown in Figure 4. This measurement randomness by Eve is rooted in Equation (5), where the four phase exponents of the matrix elements are all same. Thus, the eavesdropping randomness and

measurement indistinguishability in the shared MZI channels by Eve are sustained for $\varphi_2$—dependent network channels, resulting in the unconditional security in NC-USCKD.

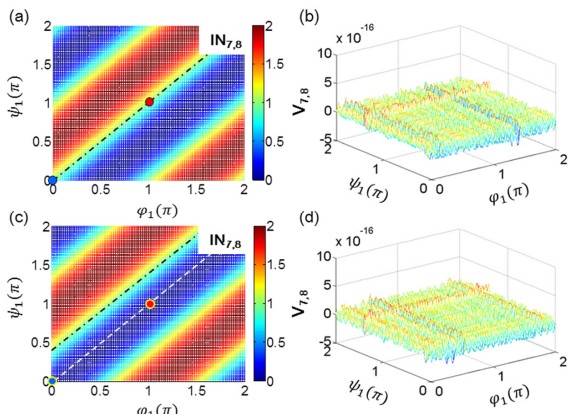

**Figure 4.** Numerical calculations of interference $IN_{7,8}$ and visibility $V_{7,8}$ for Figure 1. (**a**) $IN_{7,8}$ and (**b**) $V_{7,8}$ for $\psi_2 = \varphi_2 = 2\pi/5$. (**c**) $IN_{7,8}$ and (**d**) $V_{7,8}$ for $\psi_2 = 0$ and $\varphi_2 = 2\pi/5$. The keys are denoted by dots: $\varphi_1 = \psi_1 = 0$ (blue); $\varphi_1 = \psi_1 = \pi$ (red). $IN_{7,8} = (E_7 + E_8)(E_7 + E_8)^*$; $V_{7,8} = \left(\frac{I_8 - I_7}{I_7 + I_8}\right)$.

### 3.1. Network Initialization: Network Addressing and Authentication

In an N party attached classical network configuration through a commonly shared pair of MZI transmission channels, the network initialization includes network authentications between the two parties assigned by the corresponding address set of $\varphi_2$ and $\psi_2$. For the deterministic randomness analyzed in Figures 1–4, the network initialization between arbitrary two parties in the network is a prerequisite process to avoid any potential eavesdropping. Suppose that Alice and Bob represent any paired party in the network connected by a specific address set of $\psi_2$ and $\varphi_2$, respectively (see Figure 1). For a preparation stage, first, Alice shuffles the MZI network by randomly shifting her phase shifter $\Psi_1$ with a phase parameter $\delta(0 \leq \delta \leq 2\pi)$. Alice is now ready for scanning $\Psi_1$ for her visibility $V_A$. Second, Bob repeatedly sends the same test key encoded by his phase shifter $\Phi_1$ with $\varphi_1' \in \{0, \pi\}$ randomly. Third, Alice scans her phase shifter $\Psi_1$ until she obtains an interference fringe of the maxima. Then, Alice sets her phase basis with the $\delta$—added one: $\psi_1' \in \{\delta, \pi + \delta\}$. This modified phase set has a 50% chance of correctness due to the MZI randomness as mentioned above for Eve. The network initialization results in authentication.

Eve can also do the same as Alice does, but her chance is worse than for randomness due to $\delta$. The chance for Eve to have the same $\delta$ as Alice's is extremely low. In principle, two independent MZI systems set for Bob-Eve and Bob-Alice have a rare chance to be the same as each other, unless the input information by Bob is known to Eve, which is prohibited by definition. This small chance depends on the detector sensitivity, which is lower than one in a million in commercially available avalanche photodetectors. This sensitivity-based resolution defines the maximum number of possible addresses in the network. Of course, the network address number can be increased infinitely by using address layers, e.g., by expanding the address set $\left(\varphi_2^j, \psi_2^j\right)$ with the j hierarchy. Although Eve has luckily found the $\delta$ assigned by Alice, Eve still has 50% chance to coincide with Alice's.

The network initialization is summarized in Table 1, where the sequence number 1–4 applies for Sequence below. For this, Alice randomly resets the MZI system by modifying her phase shifter $\Psi_1$ with a new phase variable $\delta$ as mentioned above, as a preparation stage: Sequence #0. First, Bob randomly selects $\varphi \in \{\varphi_2, \varphi_2 + \pi\}$ for the light pulse $E_4$ in Figure 1 and sends it to Alice along with $E_3$ (see Figure 2): Sequence #1. Second, Alice measures $V_A$ and randomly sets her phase controller $\Psi_1$ with either $\delta$ or $\delta + \pi$ to send the reflected light to Bob: Sequence #2. Alice announces the result of $V_A$ publicly. Note that Alice never announces her phase choice either for $\psi_1$ or $\delta$. Third, Bob measures his $V_B$ and publicly announces whether Alice's measurement is correct or not: Sequence #3. Lastly,

Alice knows secretly and deterministically whether the $\delta$ is correct or wrong: Sequence #4. If it is wrong, Alice just adds a $\pi$ phase to $\delta$, otherwise keeps it as her final phase basis set of $\psi$. Table 1 is for the case of a $\pi$—phase shifted $\delta$.

- Sequence

0.  (Network preparation) Initially Alice resets the MZI network by disturbing the MZI with her phase controller $\Psi(\delta)$ and scans $\delta$ until she gets $V_A = \pm 1$ for the test bits provided by Bob. The $\delta$ is a phase variable added to her phase basis $\psi \in \{0, \pi\}$. Then, Alice gives a cue to Bob.

1.  Bob randomly selects his phase basis $\varphi \in \{0, \pi\}$, encodes his light with $\varphi$, and sends it to Alice.

2.  Alice measures $V_A$, publicly announces the result, and returns the $\varphi$-set light to Bob after encoding it with $\delta + \psi$.

3.  Bob measures $V_B$ and publicly announces whether Alice's result is correct (O) or not (X).

4.  Alice resets her phase basis $\psi \in \{0, \pi\}$ to either $\psi \in \{\delta, \pi + \delta\}$ or $\psi \in \{-\delta, \pi - \delta\}$ depending on the Bob's announcement: end of network initialization.

Eve may also perform the same network initialization of Table 1 with an arbitrary value of $\delta'$ for her phase shifter, $\Psi_e(\delta')$. As a result, Eve obtains the same pattern but with unsynchronized maxima with respect to Alice's because $\delta \neq \delta'$ due to the asymmetry of independent systems. The synchronization chance ($\delta = \delta'$) between Eve and Alice is extremely low, where the chance is decided by the detector's sensitivity as mentioned above: a commercially available detector sensitivity is very high ($>10^4$ V/W at GHz). Thus, the addressable networking with unconditional security is achieved by network initialization as shown in Table 1. The unconditional security is effective with a 50% chance (randomness) via information theory [31]. As discussed with memory-based attacks [23], Eve has no chance of eavesdropping the data. One might suggest that Eve's eavesdropping trials may shift the $V_A$ value causing an error, where the shift must be consistent owing to Eve's abilities in the coherence setup. However, a consistent $V_A$ shift to Alice does not affect the initialization process at all, otherwise, confirms Eve's intrusion. Thus, network initialization implies both network addressing and authentication between two addressees because this process completely removes the potential eavesdropping chance by Eve.

### 3.2. Key Distribution Protocol

Table 2 shows the key distribution procedure without sifting for the present NC-USCC in Figure 1. This procedure accompanies the network initialization at each order to avoid the memory-based attack, otherwise sifting is performed [23]. Below is a summary of the key distribution process: Procedure. After network initialization, Bob prepares a random key using the orthogonal bases of $\varphi_1$ and sends it to Alice via the shared MZI transmission lines. Then, Alice randomly selects the Bob-prepared one using her phase bases $\psi_1$ and set it for a raw key. Here, $\psi_1$ is modified via the network initialization in addition to the individual address $\psi_2$. Owing to the directional determinacy of MZI, both parties deterministically share the same raw key by simply reading out their visibilities ($V_A$; $V_B$). Both the identity and inversion relations in $V_B$ are used for the row keys, resulting in a nearly 100% bit rate. If bit-by-bit network initialization is not performed, then a usual sifting process is performed for a batched order based on the identity relation in $V_B$ (Section E of the Supplementary Information). In this added sifting case, the network initialization is performed for the batched order. For error corrections, both parties finally publicly announce their error bits only (red numbers), and then remove them from the row key chain. As a result, the same length of final key (m) is shared between Alice and Bob. Here, the mark X represents the discarded bit resulting from the error correction. To evaluate the error rate, Bob compares the final key chain (m) with his prepared one. Privacy amplification may be added by randomly selecting some bits in the final key chain to calculate the error bit rate. The following is the key distribution procedure for NC-USCKD (see Table 2).

- Sequence

0.  The network initialization is performed for both network addressing and authentication: see Table 1.
1.  Bob randomly selects his phase basis $\varphi_1 \in \{0, \pi\}$ to prepare a key and sends it to Alice.
2.  Bob converts the chosen basis $\varphi_1$ into a key for his key record x: $x \in \{0, 1\}$, if $\varphi = 0$, x = 0; if $\varphi = \pi$, x = 1. The $\varphi_1$ is not influenced by the network initialization process.
3.  Alice measures her visibility $V_A$ and keeps the record.
4.  Alice copies the Bob's key for her record y via MZI directionality: if $V_A = 1$, y = 0; if $V_A = -1$, y = 1; if $V_A \neq \pm 1$, y = $V_A$ (error).
5.  Alice randomly selects her phase basis $\psi_1 \in \{0, \pi\}$, encodes the return light, and sends it back to Bob. Here, the $\psi_1$ is a corrected value as a result of the network initialization process: see Table 1.
6.  Alice converts the chosen basis $\psi_1$ into a key record z: $z \in \{0, 1\}$; if $\psi_1 = 0$, z = 0; if $\psi_1 = \pi$, z = 1.
7.  Alice compares y and z for the raw key $m_A$: $m_A = (y + z) \oplus 1$ at modulus 2. If $m_A \neq \{0, 1\}$, $m_A = X$ (error).
8.  Bob measures his visibility $V_B$ and keeps the record.
9.  Bob sets the raw key $m_B$ via MZI determinacy: if $V_B = 1$, $m_B = 0$; $V_B = -1$, $m_B = 1$. If $V_B \neq \pm 1$, $m_B = X$ (error).
10. Alice and Bob publicly announce their error bits and remove them from their raw keys to set the shared final key, $\{m\}$.

## 4. Discussion

Regarding the eavesdropping discussed in Figure 4, Eve can set up the same measurement tools for both outbound and inbound eavesdropping as Alice and Bob have, respectively. Then, Eve simply reads out her visibility relying on the same MZI directionality with best chance of 50% on average. For arbitrary addressing in the N-party attached NC-USCKD, the network initialization between any arbitrary bi-parties results in network authentication. Thus, Eve's measurement-based eavesdropping for the phase-controlled round-trip MZI system of Figure 1 is worse than random, resulting in unconditionally secured cryptography, even in the classical domain. Here, the network resolution or maximum number of addresses in the network is determined by the MZI phase stability [32], where extension of the transmission distance of more than a few km range [27,28] for the shared MZI is a just technical issue [33].

*Coherence-Based Memory Attack*

The eavesdropping randomness in the MZI scheme of Figure 1 however must be consistent relative to all coherently measured bits by Eve either in phase or out of phase with Alice or Bob. This fact is critical to post-measurement attacks such as memory-based attacks because Eve can simply flip all eavesdropped bits for correction. To protect from such a classical attack, bit-by-bit network initialization (Table 1) or block-based sifting (Section C of the Supplementary Information) is necessary. In other words, the eavesdropping randomness in MZI must be bit-by-bit to satisfy unconditional security in the present scheme. Then, the maximum eavesdropping rate becomes $\eta_e = \left(\frac{1}{2}\right)^N$, where N is the key length in digits. For N = 128, $\eta_e \sim 10^{-39}$, it takes much longer than the age of the universe ($10^{35}$ s) for a brute-force attack to succeed even with the world's most powerful supercomputer, whose bit flip time is $10^{-17}$ s (see Section F of the Supplementary Information). For the random bit sequence, no efficient algorithm exists except for brute-force attacks. Owing to the coherence optics compatible with conventional optical systems, the key length of the present NC-USCKD has no practical limit due to phase-locked amplification. Thus, the unconditional security of NC-USCKD using coherent

light opens the door to potential one-time-pad cryptography in the classical domain, otherwise impossible.

## 5. Conclusions

The NC-USCKD protocol was presented, analyzed, and discussed for addressability in an N-party attached classical network, where unconditional security is based on quantum superposition between shared transmission lines in the classical regime. The key rate of NC-USCKD depends on classical optoelectronic devices such an acousto-optic or electro-optic modulators at GHz compatible with current fiber-optic communications network systems. The network initialization in the N-party-involved optical network was successfully shown for two arbitrary parties assigned by the public addresses. The number of public addresses is practically dependent on the photo-detector's sensitivity. Network initialization also resulted in authentication between the addressed two parties, where Eve's eavesdropping success rate is quadratically decreased as N linearly increases. The proposed NC-USCKD can be applied to conventional DWDM-based fiber-optic communications networks by allocating each address to each wavelength [34]. Because of the MZI robustness in phase fluctuations demonstrated in both optical fibers [24] and free space [28] for a few km ranges, the network extension to tens of km with large N is a simple technical issue with current locking technologies [27,28,33,34]. In a multi-core fiber, the MZI path length is potentially error-free due to the core-to-core proximity in a few microns [1]. The wavelength converter, optical MUX/DEMUX, and an amplifier such as EDFA are coherent devices, so a phase difference between the input and output can be locked. This fixed phase shift can also be adjusted for the desired interference fringe in a network preparation stage. For wavelength sharing/dependent network configurations, STAR, ring, or FTTH fiber optic networks are also possible.

Unconditional security in NC-USCKD by using bright coherent light was presented using addressable quantum superposition and its unitary transformation for a shared MZI system between any two arbitrary remote parties in a network. Compared with QKD protocols such as BB84 based on single photons over a single quantum channel, the unconditional security of NC-USCKD was far more superior, resulting in detection loophole-free, ultrafast and distance unlimited unconditionally secured cryptography for N parties in a network. Unlike the canonical (non-orthogonal) basis-based no-cloning theorem in QKD, the physics of unconditional security of NC-USCKD lies in the quantum superposition between paired transmission lines of the MZI channels and its unitary transformation in a round-trip scheme, resulting in deterministic randomness. To avoid potential eavesdropping, real-time network initialization was performed to protect from classical attacks such as memory-based attacks. Compared with the original point-to-point transmission scheme of USCKD, the addressability in NC-USCKD is due to the linearity of orthogonal basis expansion among N parties for N-to-N networking. Eventually, the proposed NC-USCKD can be applied to current fiber-optic communications networks with laser locking techniques as well as to future multi-core fiber networks. As a result, NC-USCKD has potential for the long-lasting goal of one-time-pad cryptography in the classical regime for artificial intelligence requiring unconditionally secured mass data communications, such as in unmanned vehicles, drones, and medical record transmission.

**Supplementary Materials:** The following supporting information can be downloaded at: https://www.mdpi.com/article/10.3390/cryptography6010004/s1.

**Funding:** This research was funded by ICT R&D program of MSIT/IITP (2021-0-01810), "Development of elemental technologies for ultra-secure quantum internet".

**Data Availability Statement:** The data presented in this study are available in article.

**Conflicts of Interest:** The author declares no conflict of interest.

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
