# Peer review of "Network-Compatible Unconditionally Secured Classical Key Distribution via Quantum Superposition-Induced Deterministic Randomness"

_cryptography, doi:10.3390/cryptography6010004_

Round 1

Reviewer 1 Report

Good paper and actual topic. Indeed, quantum key distribution is popular now. The article contains experimental data and theoretical information.
Disadvantages: 1) it is necessary to shorten the "discussion" section or rephrase. You need to write the essence. 2) What communication channel is used between network users? How does the number of photons in a pulse affect the algorithm? How does the probability of key distribution change during an attack? Have you considered open space as a communication channel?
3) I recommend that you familiarize yourself with the literature and use it: DOI: 10.3390/e23050509, DOI: 10.4018/978-1-7998-8593-1.ch015

Author Response

Response to Reviewer 1

Comment 1: Good paper and actual topic. Indeed, quantum key distribution is popular now. The article contains experimental data and theoretical information. Disadvantages: 1) it is necessary to shorten the "discussion" section or rephrase. You need to write the essence.

Response 1: The Discussion has been shortened in the revised version.

Comment 2: What communication channel is used between network users? How does the number of photons in a pulse affect the algorithm? How does the probability of key distribution change during an attack? Have you considered open space as a communication channel?

Response 2: Thank you for the valuable comment. The communication channel of USCKD is classical but composed of Mach Zehnder interferometer-type double channels. Channel tapping and copying by an eavesdropper are allowed in USCKD. However, the double channel structure randomizes for Eve to extract information according to quantum mechanics of superposition. Detailed analysis of key distribution under potential attacks is beyond the present scope (discussed elsewhere), but there is nearly no change in the key distribution rate if a smart eavesdropper is involved. This is because a smart eavesdropper may not disturb the MZI phase scheme to avoid his or her existence. Unlike quantum key distributions, NC-USCKD is less vulnerable to channel attacks due to the allowed channel copying as in any classical cryptographic methods. Regarding the open space applications, I’ve already published a journal paper: Ham, B. S. Analysis of phase noise effects in a coupled Mach-Zhender interferometer for a much stabilized free-space optical link. Sci. Rep. 2021, 11, 1900. This has been added as ref. 24 in the revised version.

Comment 3: I recommend that you familiarize yourself with the literature and use it: DOI: 10.3390/e23050509, DOI: 10.4018/978-1-7998-8593-1.ch015

Response 3: Thank you for the recommended literatures. They are added in the reference list (refs. 5, 6) of the revised version. By the way, ref. 4 has been replaced by a more suitable and direct reference for practical QKDs.

Reviewer 2 Report

The authors propose a new unconditionally secured classical key exchange method. It uses a pair of transmission channels (MZI) and a quantum superposition. The impact of Quantum computing on various areas is at the edge of security and communications research. The paper's claim, definitions, and implementation are sound and well explained. The use of related work is well done.

Author Response

Response to Reviewer 2

Comment: The authors propose a new unconditionally secured classical key exchange method. It uses a pair of transmission channels (MZI) and a quantum superposition. The impact of Quantum computing on various areas is at the edge of security and communications research. The paper's claim, definitions, and implementation are sound and well explained. The use of related work is well done.

Response: Thank you for your review report and strong recommendation of publication in Cryptography.